# In Vitro Evaluation of the Probiotic Properties and Whole Genome Sequencing of *Lacticaseibacillus rhamnosus* J3205 Isolated from Home-Made Fermented Sauce

**DOI:** 10.3390/microorganisms13071643

**Published:** 2025-07-11

**Authors:** Yiming Chen, Lingchao Ma, Weiye Chen, Yiwen Chen, Zile Cheng, Yongzhang Zhu, Min Li, Yan Zhang, Xiaokui Guo, Chang Liu

**Affiliations:** 1School of Global Health, Chinese Center for Tropical Diseases Research, Shanghai Jiao Tong University School of Medicine, Shanghai 200025, China; yimingc466@gmail.com (Y.C.); malingchao@sjtu.edu.cn (L.M.); cweiye2000@163.com (W.C.); cyw0903@sjtu.edu.cn (Y.C.); chengzile@sjtu.edu.cn (Z.C.); yzhzhu@sjtu.edu.cn (Y.Z.); minli@shsmu.edu.cn (M.L.); yan_zhang@shsmu.edu.cn (Y.Z.); 2School of Public Health, Shanghai Jiao Tong University School of Medicine, Shanghai 200025, China; 3Department of Immunology and Microbiology, Shanghai Jiao Tong University School of Medicine, Shanghai 200025, China

**Keywords:** probiotic, whole genome sequencing, proteomics, anti-inflammation, antioxidants

## Abstract

*Lacticaseibacillus rhamnosus* J3205 was isolated from traditional fermented sauces and demonstrated potential probiotic properties. The strain exhibited high tolerance to simulated saliva (93.24% survival) and gastrointestinal conditions (69.95% gastric and 50.44% intestinal survival), along with strong adhesion capacity (58.25%) to intestinal epithelial cells. Safety assessments confirmed the absence of virulence and antibiotic resistance genes. Genomic analysis revealed stress-response genes and 34 insertion sequence (IS) elements, while proteomic profiling identified *Pgk* as a key enzyme in lactic acid production and *SecY* in oxidative stress resistance. Functionally, J3205 significantly reduces pro-inflammatory cytokines (TNF-α, IL-6, IL-1β) and enhances antioxidant markers (SOD, GSH) in vitro. These results position *L. rhamnosus* J3205 as a promising candidate for gut-health foods, anti-inflammatory nutraceuticals, and oxidative-stress therapeutics, warranting further in vivo validation.

## 1. Introduction

Probiotics are defined as live microorganisms that, when ingested in adequate quantities, can confer beneficial effects on the host [1]. Lactic acid bacteria (LAB), comprising genera such as *Lacticaseibacillus*, *Bifidobacterium,* and *Streptococcus*, among others, are employed extensively in the production of food products, nutraceuticals, and pharmaceuticals. They are regarded as a substantial source of probiotics [2,3]. Probiotics have been demonstrated to possess multiple beneficial functions, including the secretion of antimicrobial substances (e.g., hydrogen peroxide and bacteriocins) to inhibit pathogenic bacteria [4], as well as the modulation of host immune responses [5,6,7,8]. Certain probiotics may improve gut health by alleviating pathogenic bacteria-induced diarrhea, enhancing intestinal barrier integrity, and reducing inflammation. Additionally, via the gut–brain axis, they can modulate neural function, potentially providing therapeutic benefits for neuropsychiatric disorders, including depression [9,10,11,12]. It is therefore one of the most important strategies for regulating gut microbiota. Given that the function of probiotics is strain-specific, the function of each strain needs to be studied in depth for better clinical application. At the same time, in addition to the existing commercialized strains, it is essential to continuously explore more microbial resources for screening new candidate probiotic strains [7,13,14,15,16].

Before candidate probiotic strain can be considered for human application, it must undergo rigorous evaluation to ensure safety, functionality, and viability. Safety is paramount, requiring strains to be non-pathogenic, non-hemolytic, and sensitive to clinically relevant antibiotics to prevent potential health risks and antibiotic resistance transmission. Additionally, comprehensive genomic screening should confirm the absence of virulence factors. Functionality assessment includes verifying strain-specific beneficial effects such as antimicrobial activity against pathogens, immunomodulatory capabilities, and the ability to adhere to intestinal epithelial cells. Finally, viability must be demonstrated through tolerance to gastrointestinal conditions, including gastric acid and bile salts, as well as stability during processing and storage. Only strains that satisfy all these critical criteria can be further developed and moved towards potential clinical applications [14,15,17].

This study aims to isolate and identify potential probiotic strains from traditional family-fermented sauces, and to evaluate their benefits and safety for human applications. The commercially available probiotic strain *L. rhamnosus* GG (LGG, ATCC 53103) was used as a control. The results revealed that this strain possesses excellent potential probiotic properties, particularly in alleviating oxidative stress and inflammation, suggesting its potential therapeutic value in treating gut-related and metabolic disorders.

## 2. Materials and Methods

### 2.1. Identification of the Bacterial Strains

The candidate probiotic was isolated from a traditional homemade sauce originally fermented from soybeans. A 2.5 g sample of the sauce was dissolved in 10 mL of 1 × Phosphate Buffered Saline (PBS) (Jizhi Biochemical Co., Ltd., Nanjing, China) buffer, serially diluted to 10^−5^, and evenly spread on de Man, Rogosa, and Sharpe (MRS) (Qingdao Haibo Biotechnology Co., Ltd., Qingdao, China) agar plates, followed by anaerobic incubation for 48 h. The bacteria were purified through two successive generations of culture. After homogenizing 2.5 g of the sauce sample in 10 mL of 1 × PBS buffer, it was serially diluted to 10^−5^ and plated on MRS agar. Following 48 h of anaerobic incubation at 37 °C, single colonies were purified through two successive subcultures to ensure strain consistency. Colonies were selected and identified using matrix-assisted laser desorption ionization time-of-flight mass spectrometry (MALDI-TOF MS). Identification was based on the mass-to-charge ratio (m/z) and the degree of relative intensity matching of the mass spectrometry peaks, with the identification result determined by the matching score (Score). A Score ≥ 9 was established as a reliable identification. Additionally, Average Nucleotide Identity (ANI) analysis based on whole genome sequencing was conducted for the confirmation of identification.

### 2.2. Whole Genome Sequencing and Assembly

Whole genome sequencing and assembly were completed by Shanghai Personal Biotechnology Co., Ltd. (Shanghai, China). The OrthoANI tool (https://www.ezbiocloud.net/tools/orthoani v0.90, accessed on 2 August 2024) was utilized to analyze the Average Nucleotide Identity (ANI) [18]. Genes were subjected to BLAST searches against the CAZy, KEGG, COG, and GO classification databases of strain J3205. The hmmscan software (v3.2.1) was employed to predict the presence of CAZy enzyme-like genes within the genomic sequence [19]. Open Reading Frame (ORF) sequences longer than 80 amino acids were selected, with an E-value threshold set at 1 × 10^5^ Additionally, amino acid sequences exceeding 30% of the database length were included, while sequences shorter than 80 amino acids were also considered, with an E-value threshold of 1 × 10^5^ and amino acid sequences longer than 30% of the database length. The eggNOG database encompasses functional classifications derived from the original COG/KOG (COG for prokaryotes; KOG for eukaryotes). In bacterial cases, a protein sequence is annotated to a specific eggNOG (COG) through comparative analysis, with COGs classified into twenty-five functional categories [20]. The sequences of the encoded proteins were aligned with those from the eggNOG (COG) database using DIAMOND blastp (v2.0.14), with a sequence alignment threshold of 1 × 10^6^. The eggNOG number corresponding to the best hits was assigned to the relevant protein-coding gene [21]. Gene Ontology (GO) annotations of protein-coding genes were performed using BLAST2GO software (v1.0) [22]. Furthermore, cgview was employed to visualize the genome of strain J3205 [23]. Transfer RNA (tRNA) and ribosomal RNA (rRNA) were identified using tRNAscan-SE and RNAmmer, respectively.

#### 2.2.1. Genomic Stability Analysis of J3205

The prophage, IS elements and CRISPR-Cas system associated with genome stability were predicted online using PHASTEST (https://phastest.ca/) (v3.0, accessed on 29 June 2025), IS finder (https://isfinder.biotoul.fr/) (Insertion sequences were analyzed using the ISfinder database (last updated: 6 June 2025, accessed on 29 June 2025), and CRISPRCas Finder (https://crisprcas.i2bc.paris-saclay.fr/) (CRISPR-Cas++ v1.1.2, I2BC, accessed on 29 June 2025)), respectively [24,25,26].

#### 2.2.2. Prediction of J3205 Virulence Factors and Antibiotic Resistance Genes

Antibiotic resistance genes were predicted using the Comprehensive Antibiotic Resistance Database (CARD, https://card.mcmaster.ca/) (v3.2.6, accessed on 7 April 2025), under default settings (perfect/strict option). Virulence genes were predicted using the Virulence Factor Database (VFDB, http://www.mgc.ac.cn/VFs/) (last updated: 7 April 2025, accessed on 7 April 2025), and the parameter settings were as follows: E-value = 1 × 10^−^^5^; coverage >70%; and identity >70%.

#### 2.2.3. Comparative Analysis of Secretary Proteomics of Strains J3205 and LGG

The extraction and characterization of secretome proteins were entrusted to Shanghai Ouyi Biological Co., Shanghai, China. The brief steps were as follows: culture supernatants of two strains were centrifuged (12,000× *g*, 4 °C, 10 min) to remove cell debris, and then concentrated proteins were extracted using the BCA kit method. This was followed by trypsin digestion and analyzed using liquid chromatography-tandem mass spectrometry (LC-MS/MS). Raw data were combined with all mass spectrometry data by the software Spectronaut Pulsar 18.7 (Biognosys) and compared to a database for protein identification and quantification. The screening criteria for differentially expressed proteins were fold change >2.0 or <0.5 and *p*-value < 0.05.

### 2.3. Artificial Saliva, Gastric, and Intestinal Fluid Tolerance Experiment

Stability tests using Artificial Saliva Fluid (ASF), Artificial Gastric Fluid (AGF), and Artificial Intestinal Fluid (AIF) were conducted as previously described [27,28]. The bacterial suspension was centrifuged at 6000× *g* for 5 min and then incubated with 100 mL of artificial saliva fluid (25 mM KH_2_PO_4_, 24 mM Na_2_HPO_4_, 150 mM KHCO_3_, 100 mM NaCl, 1.5 mM MgCl_2_, 25 mM citric acid, 15 mM CaCl_2_, pH 7.0) (Macklin Biochemical Technology Co., Ltd., Shanghai, China), adjusted to 10^8^ CFU/mL, for 3 h. Subsequently, the bacterial suspension was resuspended in artificial gastric fluid (2 g/L NaCl, 3 g/L pepsin, pH 2.0), again adjusted to 10^8^ CFU/mL, and incubated for another 3 h. Following this, the bacteria were resuspended in artificial intestinal fluid (0.3% bile salt, 1 g/L trypsin in PBS, pH 8.0) for 3 h. The bacterial count was determined hourly using the plate counting method, with the bacteria incubated anaerobically at 37 °C. After 3 h of incubation, the bacteria were resuspended in artificial intestinal juice (0.3% bile salts, 1 g/L pancreatin, pH 8.0) and treated for an additional 3 h. Bacterial counts were recorded every one hour.(1)$$Survival\;rate\;\%=\frac{lgN_{t}(CFU)}{lgN_{0}(CFU)}\;\times 100\%$$
in which N_t_ is equals to the number of viable bacteria after treatment and N_0_ is the number of initial live bacteria.

### 2.4. Antagonism of Pathogenic Bacteria

The antibacterial activity of J3205 supernatant against foodborne pathogens was evaluated using the agar well diffusion assay. Indicator strains included *Listeria monocytogenes* 10430s, *Staphylococcus aureus* ATCC 6538, *Escherichia coli* ATCC 25922, *Salmonella typhimurium* 14028, and *Clostridioides difficile* ATCC 43255. The logarithmic phase of the pathogen was adjusted to 10^8^ CFU/mL, and 200 μL of the bacterial solution was taken to the LB plate and spread evenly, a 6 mm well was made in the plate. An amount of 100 μL of J3205 supernatants were taken and injected into the well, and MRS liquid medium was used as a negative control. The plate was transferred to a constant temperature and humidity incubator and cultured at 37 °C. After 24 h of incubation, the diameter of the inhibition zone was measured accurately with a vernier caliper wherever the inhibition zone was observed. In addition, BAGEL4 was used to predict the types of bacteriocins (http://bagel4.molgenrug.nl/) (v1.2 accessed on 14 May 2025) [29].

### 2.5. Hydrophobicity Ability Test

Overnight cultures were centrifuged at 5000× *g* for 10 min at 4 °C. The cell pellets were washed twice with sterile phosphate-buffered saline (PBS, pH 7.2) and resuspended in PBS to an optical density of 0.6 ± 0.05 at 600 nm (A_0_). An amount of 2 mL of bacterial suspension was mixed vigorously with an equal volume of chloroform for 2 min and allowed to stand at room temperature for 30 min. The aqueous phase was carefully collected, and its optical density at 600 nm (A_1_) was measured against PBS. Hydrophobicity percentage was calculated as(2)$$Hydrophobicity\;rate\;\%=\frac{A_{0}-A_{1}}{A_{0}}\times 100\%$$

### 2.6. Auto-Aggregation Ability Test

The preparation of the bacterial solution is the same as the hydrophobicity experiment, by taking 4 mL of the bacterial suspension and aspirating 1 mL of the upper layer; the value of OD600, A_0_, was measured. It was left to stand at room temperature for 12 h and then 1 mL of the upper layer was carefully aspirated; the value of OD600, A_1_, was measured. The formula is as follows:(3)$$Auto-aggregation\;rate\;\%=\frac{A_{0}-A_{1}}{A_{0}}\;\;\times 100\%$$

### 2.7. Antibiotic Susceptibility Test

The antimicrobial susceptibility of the strain to seven antibiotics, including Streptomycin, Gentamicin, Clindamycin, Kanamycin, Tetracycline, Ampicillin, and Erythromycin, was evaluated using the E-test assay. The single colony was picked up into 0.85% NaCl solution and adjusted to McFarland Turbidity at 0.5, and then evenly spread onto MRS agar medium, then left on the plate to dry. Sterile E-test strips were attached to the plate. Antibiotic criteria were based on EFSA 2012 [30].

### 2.8. Hemolytic Test

The isolated strains were inoculated on Columbia blood agar plates and incubated anaerobically at 37 °C for 24 h [31]. *S. aureus* ATCC 6538 was used as a positive control.

### 2.9. Viability of L. rhamnosus J3205 During Storage

The bacterial strain was inoculated into MRS liquid medium and incubated anaerobically at 37 °C for 18–24 h (late logarithmic phase, OD600 ≈ 1.0–1.2). Bacterial pellets were harvested by centrifugation (4 °C, 6000× *g*, 10 min), washed twice with sterile PBS, and resuspended in 20% (*w*/*v*) skim milk powder at a 1:1 ratio (bacterial suspension to skim milk). Aliquots (300 μL) were dispensed into sterile glass bottles (four bottles per strain, corresponding to storage days 0, 7, 14, and 21). The samples were flash-frozen in liquid nitrogen and lyophilized. Viable cell counts were determined on days 0, 7, 14, and 21 by standard plate counting on MRS agar.

### 2.10. Cell Culture

Human colorectal adenocarcinoma cell line (Caco-2), intestinal epithelioid cell line No. 6), and murine macrophages (Raw 264.7) were cultured in complete medium (Dulbecco’s modified eagle medium) supplemented with 10% (*v*/*v*) heat-inactivated fetal bovine serum (FBS) and 1% penicillin–streptomycin (P/S) (all reagents were from Gibco, Thermo Fisher Scientific, Shanghai, China). The Caco-2 cells were incubated in a humidified incubator with 5% CO_2_ at 37 °C for all experiments.

#### 2.10.1. Cell Adhesion Assay

The adhesion of bacteria to Caco-2 cells were assessed [32]. An in vitro adhesion assay was performed using Caco-2 cells cultured in complete medium at 37 °C with 5% CO_2_ in a humidified incubator. After 2 passages, cells were seeded into 6-well plates and incubated at 1 × 10^6^/mL for 24 h. Cells were then washed twice with PBS. *L. rhamnosus* J3205, was centrifuged at 6000× *g* for 10 min, and washed three times with sterile PBS, then resuspended in the complete medium at a concentration of 1 × 10^8^ CFU/mL. Caco-2 cells were washed with sterile PBS and 1 mL of bacterial broth (1 × 10^8^ CFU/mL), added to each well, and then co-cultured with Caco-2 cells at 37 °C for 5 h. Then the cells were washed 5 times with sterile PBS. An 0.3% (*v*/*v*) Triton X-100 in 1 mL of sterile PBS was added to each well and incubated at 4 °C for 30 min. After obtaining the cells, they were transferred to sterilized 1.5 mL test tubes, centrifuged at 6000× *g* for 10 min, and washed twice with sterile PBS. Finally, the cells were resuspended in sterile PBS and serially aseptically diluted for inoculation of MRS agar plates and incubated at 37 °C for 48 h to assess the adhesion capacity.(4)$$Adhesion\;rate\;\%=\frac{lgN_{t}(CFU)}{lgN_{0}(CFU)}\times 100\%$$in which N_1_ represents the total bacterial counts adhered to Caco-2 cells, N_0_ equals to 10^8^ CFU/mL.

#### 2.10.2. Cell Viability Assay

The cytotoxicity of cell proliferation was assessed using the CCK8 assay. J3205 and LGG were cultured anaerobically in MRS broth for 24 h, and the supernatant was collected by centrifugation and filtration. IEC-6 cells were seeded into 96-well plates at 5 × 10^4^ cells/well. After overnight incubation, cells were serum-starved for 12 h with DMEM basic medium, followed by treatment with 20% sterile supernatant (experimental group), 20% fresh MRS (control group), or DMEM (blank control) for 24 h. Cells were washed and CCK8 reagent (1:10 in DMEM) was added for 1 h. Absorbance at 450 nm was measured.(5)$$Cell\;viability\%=\frac{Experiment\;cell\;OD-Blank\;OD}{Control\;cell\;OD-Blank\;OD}\times 100\%$$

#### 2.10.3. Cytokines Measurement

Raw 264.7 cells were cultured in the complete medium. Cells were seeded at 1 × 10^6^ cells/mL in 6-well plates for 24 h. J3205 and LGG CFS (OD600 of 1.0 units) were collected by centrifugation and 0.22-μm filtration. A 20% concentration of CFS of J3205 and LGG diluted in the complete medium was added to each well for 2 h, followed by 2 μg/mL LPS for 16 h. The control group was treated with complete medium for 18 h. Total RNA was extracted with Trizol reagent, and mRNA expression of IL-10, IL-6, IL-1β, and TNF-α was quantified by RT-qPCR with β-actin as the reference gene. Primer sequences are listed in Table 1. 

#### 2.10.4. Determination of Antioxidant Enzymes

Cell supernatant processing consistent with 2.7.3, Cells were then collected to measure SOD, GSH, GSSG (Beyotime Biotechnology kits, Beyotime, Shanghai, China), and H_2_O_2_ levels (Beyotime Biotechnology kit, Beyotime, Shanghai, China) according to the manufacturer’s instructions.

### 2.11. Statistical Analysis

All tests were conducted three times, and the results are shown as mean ± standard deviation. Statistical analysis was performed with GraphPad Prism (GraphPad Prism 10.0). Statistical significance between groups was evaluated using unpaired *t* test, and statistical significance among groups was evaluated by Ordinary one-way ANOVA, with *p* < 0.05 considered statistically significant.

## 3. Results

### 3.1. Molecular Identification of the Isolated Strain

The ANI analysis of J3205 is shown in Figure 1. Based on the ANI analysis, J3205 showed high identity (≥99%) to the *L. rhamnosus* strain GCA_002158925.1 deposited in the National Center for Biotechnology Information (NCBI). In addition, MALDI-TOF MS had the same result as the ANI analysis. J3205 was confirmed as *Lacticaseibacillus rhamnosus.*

### 3.2. Genomic Characteristics of L. rhamnosus J3205

*L. rhamnosus* J3205 contained one circular chromosome, with the size of 2,882,186 bp, without plasmid, and the average GC% was 46.77% (Table 2). The complete circular genome map of strain J3205 is shown in Appendix A. Three CRISPR sequences and one Cas sequence were detected in the chromosome. However, only one of them has high evidence level at 4 (Appendix A). Two complete prophages and 34 IS elements were found in the chromosome (Appendix A). Using the bacteriocin online software BAGEL4 (http://bagel4.molgenrug.nl/) (v1.2, accessed on 14 May 2025) to mine genes for potential bacteriocin gene clusters, the matching bacteriocin was enterocin X β chain (enterocin_X_chain_beta), a Class IIb two-peptide bacteriocin, which is usually produced by lactic acid bacteria (e.g., the genus Enterococcus) (Appendix A).

The CAZy function annotation indicated 16 Carbohydrate Esterases (CE), 44 Glycoside Hydrolases (GH), and 29 Glycosyl Transferases (GT) (Figure 2A). The COG (Clusters of Orthologous Groups of proteins) classification showed that 2265 genes were annotated, and most of them are related with carbohydrate transport and metabolism (Figure 2B). The KEGG analysis covered 44 pathways with 1348 genes documented, in which metabolism and brite hierarchies were dominant (Figure 3A). There were 1900 GO functions found (Figure 3B), which are mainly associated with biological processes, cellular components, and molecular functions.

### 3.3. Genes Related to Antibiotic Resistance and Virulence

No predicted antibiotic resistance genes were found in the genome of *L.rhamnosus* J3205. Five predicted virulence factors were found in the genome of *L.rhamnosus* J3205 (Table 3). All the virulence factors are not aggressive. *tuf* and *groEL* are related to bacterial adhesion, *rfbB* is an immune modulator, *galF* is related to gluconeogenesis, and *arlR* is associated with response regulator.

### 3.4. Secretion Proteomics of Supernatants from J3205

Differential protein screening revealed that J3205 had 15 significantly upregulated and 13 significantly downregulated proteins compared to LGG (log2FoldChange <−1, *p* < 0.05) (Figure 4A,B). The hierarchical clustering analysis revealed that the expressions of the *Pgk*, *SecY*, *CarB,* and *RpmJ + RplE* of strain J3205 were upregulated (Figure 4C). The GO enrichment analysis indicated that J3205 upregulated genes were mainly concentrated in biosynthesis, cellular component, and molecular function (Figure 4D); the InterPro analysis demonstrated that the upregulated gene expression of J3205 includes ribosomal subunit, DNA-binding protein, and *Amp* receptor, etc (Figure 4E). The KEGG enrichment analysis indicated that J3205 showed enriched pathways in metabolism, genetic information processing, environmental information processing, human diseases, and cellular processes (Figure 4F).

### 3.5. Probiotic Properties of L. rhamnosus J3205

#### 3.5.1. Antagonistic Activity

An obvious antimicrobial activity against *C. difficile* ATCC 43255 and *L. monocytogenes* ATCC 19115 was observed in *L. rhamnosus* J3205 compared to LGG, and the inhibition zone reached 21.32 ± 0.94 mm and 14.87 ± 0.59 mm, respectively (Figure 5).

#### 3.5.2. Growth Curve, Acid Production, and Hemolytic Activity

The growth curve showed that J3205 entered the exponential phase after 3 h of incubation and reached the stationary phase after 18 h. The OD600 was about 1.5, which was comparable to that of the control strain LGG (Figure 6A).

Acid-producing ability of the J3205 was higher than LGG, with the final pH at 5.24 (Figure 6B). J3205 and LGG showed no hemolytic zone on the blood agar plates (Appendix A).

#### 3.5.3. Artificial Saliva Fluid, Gastric Fluid, and Intestinal Fluid Tolerance

Notably, in the ASF environment, J3205 and LGG exhibited high survival rate, at 93.24% and 93.36%, respectively in 3 h. In the AGF environment, J3205 and LGG showed the survival rate of 69.95% and 75.69%, respectively. In the AIF environment, J3205 and LGG had survival rates up to 50.44% and 66.51%, respectively (Figure 7).

#### 3.5.4. Antibiotic Susceptibility

To determine the safety of the strains, the antibiotic susceptibility of J3205 and LGG strains was evaluated according to the European Food Safety Authority (EFSA) 2012 guidelines, and the results are listed in Table 4. Two strains were susceptible to Tetracycline, Ampicillin, and Erythromycin. Both strains were resistant to kanamycin.

#### 3.5.5. Storage Viability of J3205

The stability of bacterial strains during storage was evaluated over a 21-day period. Strain J3205 demonstrated superior storage viability, retaining an average of 65.57% of its initial concentration by Day 21. In contrast, LGG showed a lower retention rate of 64.43% under identical conditions (Figure 8).

#### 3.5.6. Hydrophobicity Rate, Auto-Aggregation Rate, and Adhesion Rate of J3205

Auto-aggregation ability was found to be higher in *L. rhamnosus* J3205 (79.33%) than in LGG (63.20%) (Figure 9A). Hydrophobicity was observed at 61.90% for LGG and 53.60% for J3205 (Figure 9B). Adhesion ability to Caco-2 cells was exhibited well by *L. rhamnosus* J3205 (adhesion rate of 58.25%), with no significant differences compared to the LGG strain, for which an adhesion rate of 54.31% was recorded (LGG: positive control) (Figure 9C).

#### 3.5.7. No Cytotoxicity of J3205 CFS on IEC-6 Cell

Under 5% CFS concentration, LGG and J3205 CFS had proliferative effects on IEC-6 cell (Figure 10A). Under 5%, 10%, and 20% CFS concentration, the inhibition of J3205 and LGG CFS on IEC-6 cell was not significant compared with the control group (Figure 10B,C). Overall, the bacterial secretion supernatants were safe for IEC-6 cells under 5%, 10%, and 20% concentrations.

#### 3.5.8. Anti-Inflammatory Effect of J3205 CFS on LPS-Treated Raw 264.7

LPS treatment significantly stimulated the mRNA expression of pro-inflammatory cytokines, including TNF-α, IL-6, and IL-1β. However, cells treated with *L. rhamnosus* J3205 and *L. rhamnosus* LGG CFS showed reduced expression levels of TNF-α, IL-6, and IL-1β (Figure 11A,B,D), whereas IL-10, anti-inflammatory cytokine was upregulated (Figure 11C).

#### 3.5.9. Antioxidant of J3205 CFS on LPS-Treated Raw 264.7

Compared with untreated cells, supplementation with LPS reduced the activity of antioxidant enzymes. LPS induced Raw 264.7 macrophages showed a decreased activity of superoxide dismutase (SOD) and glutathione (GSH), and an increased amount of H_2_O_2_ at the second hour. J3205 and LGG treatment significantly increased the activity of superoxide dismutase (SOD) and GSH compared to cells treated with LPS, and significantly decreased the amount of H_2_O_2_ at the second hour (Figure 12A–C).

## 4. Discussion

In this study, different centrifugal forces were used to separate bacterial pellets and supernatants (6000× *g* for bacterial pellet collection and 12,000× *g* for supernatant clarification). Low-speed centrifugation was used to preserve bacteria and prevent damage, while cells in the supernatant were typically collected by high-speed centrifugation as previously described [33].

Numerous studies have shown that probiotics can colonize and survive in the gastrointestinal tract of humans and animals, by maintaining microbiota balance, promoting digestion and metabolic processes, and regulating immune responses. Therefore, probiotics can enhance host immunity and improve human and animal health [34,35]. However, probiotics should meet safety and functionality requirements before being introduced to the market [13,15,36]. Home-made sauces, such as traditional fermented foods, have a long and safe history of use and serve as a superior source of probiotic isolates. The present study concentrated on *L. rhamnosus* J3205 derived from home-made fermented sauce, and conducted in vitro tests and whole-genome sequencing in order to elucidate its potential biological functions and characteristics. The whole-genome sequencing of J3205 revealed functional annotations associated with immune system processes, biological adhesion, metabolic pathways, and galactose metabolism. These genomic insights, corroborated by CAZy, GO, KEGG, and COG analyses, further validate the potential of J3205 for applications in the food and pharmaceutical industries [37]. However, despite the promising nature of these in vitro and genomic findings, additional in vivo studies are crucial to confirm its efficacy and safety prior to commercial application. The predicted bacteriocin enterocin X β may be linked to the inhibition of pathogenic bacterial growth [38].

The stability of probiotic genomes is closely related to the risk of horizontal transfer of antibiotic resistance genes and virulence genes. Mobile genetic elements, including plasmids, prophages, and insertion sequences, play a significant role in horizontal gene transfer in bacteria. The absence of plasmids in the genome of *L. rhamnosus* J3205 indicates a reduced risk of exogenous DNA interference [39]. The chromosome harbors two complete prophage genes and a highly evidenced CRISPR system. Although six insertion sequence (IS) elements were identified in IS finder, highly pathogenic IS elements such as IS26, IS1, and IS911 were not detected. Furthermore, the genome lacks resistance genes, and none of the virulence genes have been shown to be associated with known pathogenic effects [40]. Therefore, at the genomic level, the strain is considered safe. At the phenotypic level, J3205 exhibits a favorable profile, demonstrating susceptibility to commonly used antibiotics such as ampicillin, tetracycline, and erythromycin, while showing natural resistance to kanamycin, consistent with typical *Lacticaseibacillus* strains [35,41,42]. Additionally, the absence of hemolytic activity further confirms its safety for human and animal consumption. These findings align with established safety criteria for probiotics and reinforce the suitability of J3205 for commercial applications.

Proteomic analysis revealed that *L. rhamnosus* J3205 exhibits unique molecular adaptations that may confer physiological advantages over the widely used commercial strain LGG. The upregulation of phosphoglycerate kinase (*Pgk*), a key glycolytic enzyme, suggests an enhanced glycolytic flux in J3205, potentially leading to increased lactate production. This metabolic advantage could enhance its competitiveness in ecological niche colonization that necessitates rapid acidification, such as in gut microbiota or industrial fermentation processes [43]. Similarly, the overexpression of *SecY* may be associated with its response to oxidative stress. The Sec system is upregulated during stress to maintain protein homeostasis [44], indicating the potential antibacterial and antioxidant capacities of J3205.

In terms of phenotypic characteristics, this study highlights the significant potential of *L. rhamnosus* J3205 as a promising probiotic strain, supported by a comparison of its functional properties and safety profile with those of the commercial strain LGG. Notably, J3205 produced larger inhibition zones against *L. monocytogenes* and *C. difficile* (Figure 5). This enhanced antimicrobial capacity may be due to J3205’s higher acid production (Figure 6B) [45,46]. This acidification capability creates an unfavorable environment for pathogenic bacteria, aligning with previous studies that emphasize the critical role of acid production in the antimicrobial activity of *Lacticaseibacillus* strains [47,48,49,50,51,52,53]. Additionally, secretory proteomics of the supernatant revealed a significantly higher expression level of the CarB protein in *L. rhamnosus* J3205 compared to LGG. The CarB protein functions as a key catalytic enzyme in pyrimidine biosynthesis and given the established relationship between pyrimidine metabolism and bacteriocin production, these findings suggest that the enhanced antibacterial activity observed in J3205 culture supernatant may be mechanistically linked to this metabolic pathway [54]. Notably, the observed correlation between increasing lactate concentrations and the decline in pathogen viability further underscores the strain’s potent antimicrobial mechanism, which warrants further investigation to elucidate specific pathways. The 21-day storage stability indicates that both LGG and J3205 are highly resistant to degradation caused by storage in a refrigerated environment at 4 °C. Additionally, a key criterion for probiotic efficacy is the ability to survive and colonize the gastrointestinal tract. While both strains showed excellent gastric acid resistance, J3205 exhibited superior auto-aggregation and comparable hydrophobicity (Figure 9A,B). However, while these in vitro simulations provide valuable preliminary insights, the complex and dynamic nature of the gastrointestinal environment necessitates further validation through animal models and clinical trials to fully assess colonization efficiency and functional benefits in vivo.

J3205 also exhibited moderate hydrophobicity and auto-aggregation ability, both of which are critical for initial adhesion to the intestinal mucosa and the competitive exclusion of pathogens [55]. Its ability to adhere to Caco-2 cells, which serve as a model for intestinal epithelial cells, further supports its potential for long-term colonization and direct interaction with host cells. This strong adhesion capability is essential for the exertion of probiotic functions, such as regulating intestinal barrier integrity, secreting beneficial metabolites, and inhibiting the adhesion of harmful bacteria [56]. In LPS-stimulated macrophages, J3205 demonstrated more potent anti-inflammatory effects than LGG, evidenced by a greater downregulation of TNF-α and IL-6 (Figure 11A,D). Notably, The J3205 treatment resulted in a unique enhancement of IL-10 expression, a response that was not observed with LGG. (Figure 9C). Antioxidant assays revealed J3205’s superior capacity to elevate SOD activity and reduce H_2_O_2_ levels (Figure 12A,C), suggesting a stronger ability to mitigate oxidative stress. These properties are crucial for alleviating oxidative stress and inflammation, thereby protecting intestinal cells and preventing conditions such as inflammatory bowel disease (IBD). The potential involvement of the Nrf2 pathway in mediating these effects presents a promising avenue for further mechanistic exploration [57,58].

Compared to the well-known probiotic strain LGG, J3205 distinguishes itself through its favorable combination of antibacterial, anti-inflammatory, and antioxidant properties, along with its robust adhesion and colonization potential. These characteristics, in conjunction with its established safety profile, position J3205 as a competitive candidate to mitigate the current reliance on imported probiotic strains within the food market. The development of J3205 not only enriches the domestic probiotic resource pool but also contributes to the advancement of probiotic industry, which has historically encountered challenges related to limited strain resources and technological maturity.

## 5. Conclusions

In conclusion, *L. rhamnosus* J3205 represents an advancement in probiotic research, offering a scientifically validated and functionally robust alternative to imported strains. Its comprehensive safety and functional assessments, supported by genomic insights, underscore its potential to enhance human and animal health. Future studies focusing on in vivo validation and mechanistic exploration will further solidify its position as a leading probiotic strain for commercial applications.

## Figures and Tables

**Figure 1 microorganisms-13-01643-f001:**
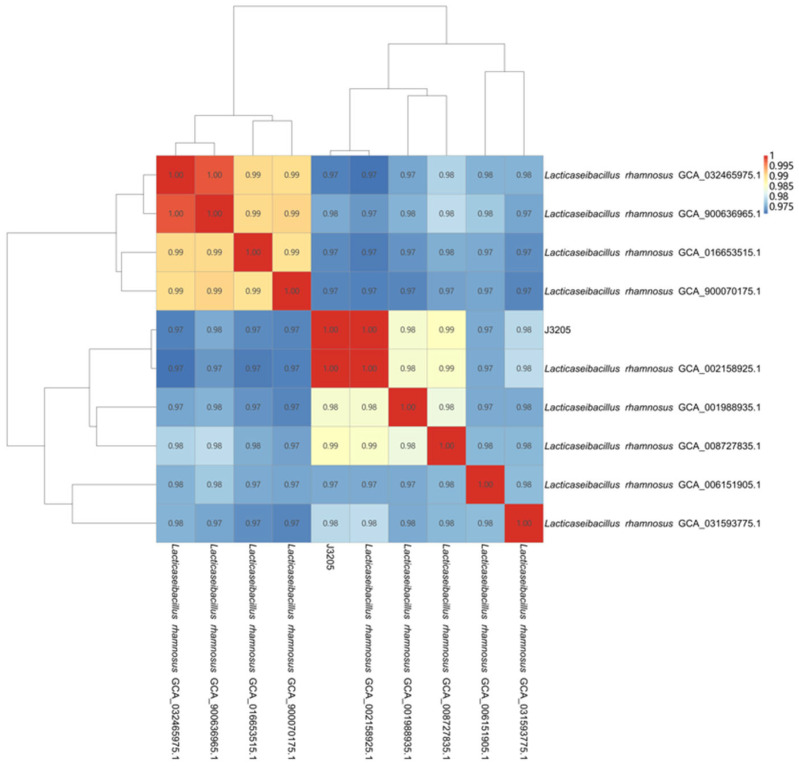
Average Nucleotide Identity (ANI) analysis of *L. rhamnosus* J3205. Whole-genome comparison of J3205 with reference strains using OrthoANI. The circular heatmap displays pairwise ANI values (%), with J3205 showing 99.9% identity to *Lacticaseibacillus rhamnosus* GCA_002158925.1 (highlighted in red). Scale: 95–100% identity. Analysis performed with OrthoANI tool (https://www.ezbiocloud.net/tools/orthoani v0.90, accessed on 2 August 2024) [20].

**Figure 2 microorganisms-13-01643-f002:**
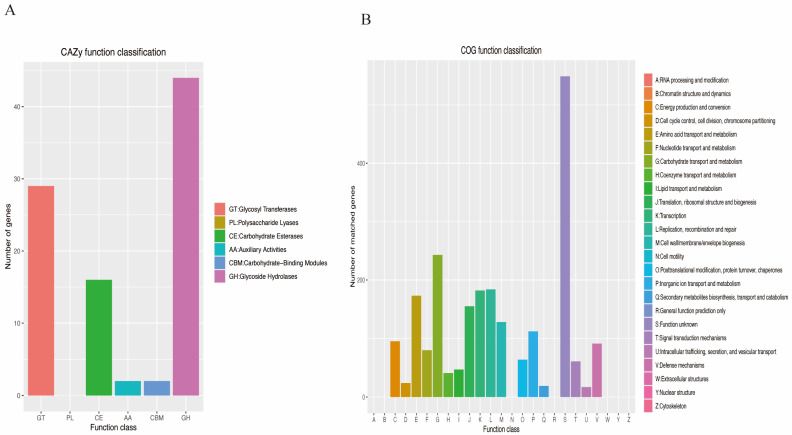
Functional annotation of *L. rhamnosus* J3205 genome. (**A**) CAZy (Carbohydrate-Active Enzymes) annotation showing distribution of enzyme classes: Glycoside Hydrolases (GH, 44 genes), Glycosyl Transferases (GT, 29 genes), and Carbohydrate Esterases (CE, 16 genes). (**B**) COG (Clusters of Orthologous Groups) classification of 2265 annotated genes, with predominant functions in carbohydrate transport/metabolism (Category G) and amino acid metabolism (Category E). Color bars represent gene counts per category.

**Figure 3 microorganisms-13-01643-f003:**
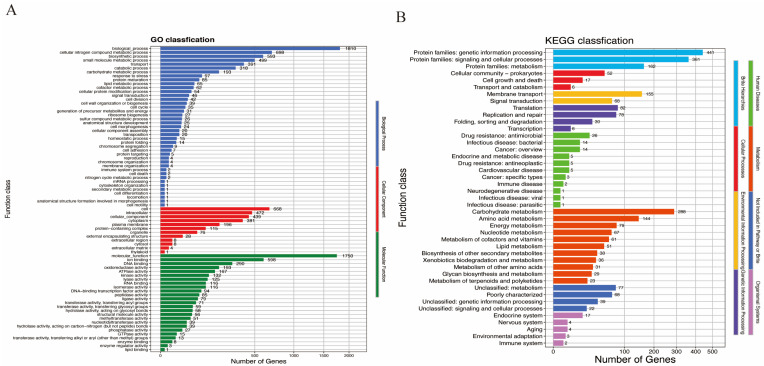
Pathway and functional classification of J3205 genome. (**A**) Gene Ontology (GO) terms categorized into biological processes, cellular components, and molecular functions. (**B**) KEGG pathway annotation of 1348 genes, dominated by metabolic pathways and biosynthesis of secondary metabolites.

**Figure 4 microorganisms-13-01643-f004:**
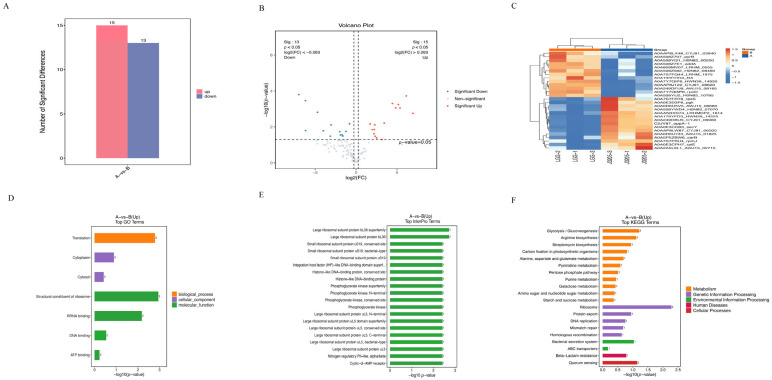
Comparative proteomic analysis between supernatants from the experimental group J3205 and the control group LGG. (**A**) Number of significant differential protein expression between J3205 and LGG in quantitative histogram. (**B**) Number of significant differential protein expression between J3205 and LGG in quantitative volcano plot. (**C**) Clusters analysis of top differential protein expression between J3205 and LGG. (**D**) GO enrichment analysis of differential proteins between J3205 and LGG. (**E**) InterPro enrichment analysis of differential proteins between J3205 and LGG. (**F**) KEGG enrichment analysis of differential proteins between J3205 and LGG.

**Figure 5 microorganisms-13-01643-f005:**
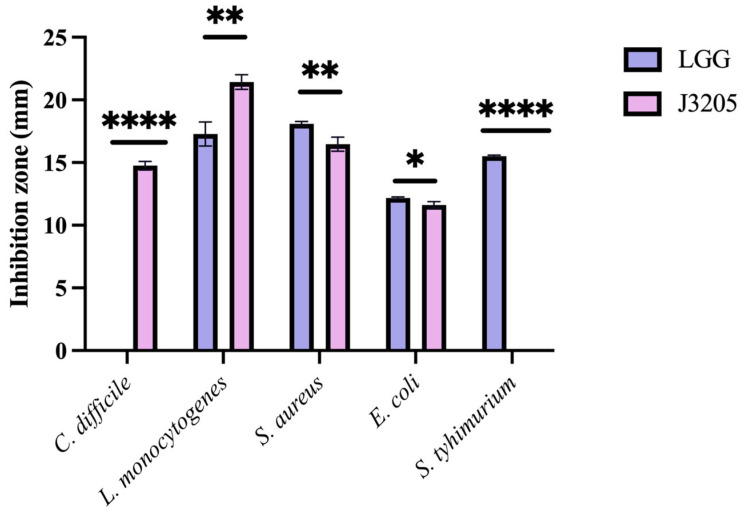
Antimicrobial activity of *L. rhamnosus* J3205 and LGG against foodborne pathogens. Inhibition zones (mm) of J3205 and LGG supernatants against *L. monocytogenes* ATCC 19115, *C. difficile* ATCC 43255, *S. aureus* ATCC 6538, and *S. typhimurium* 14028, measured by agar well diffusion assay. Data represent mean ± SD (*n* = 3). Significance vs. LGG: * *p* < 0.05, ** *p* < 0.01, **** *p* < 0.0001. LGG showed no inhibition against *C. difficile*.

**Figure 6 microorganisms-13-01643-f006:**
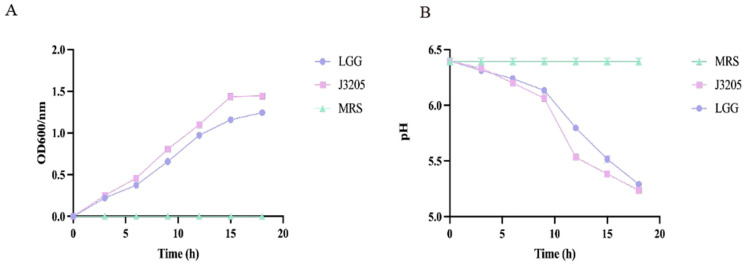
Growth rates and acid production of *L. rhamnosus* J3205. (**A**) Growth curves of J3205 and LGG in MRS broth at 37 °C (OD600 measured hourly). (**B**) pH reduction during 18 h fermentation. Data are shown as mean ± SD (*n* = 3).

**Figure 7 microorganisms-13-01643-f007:**
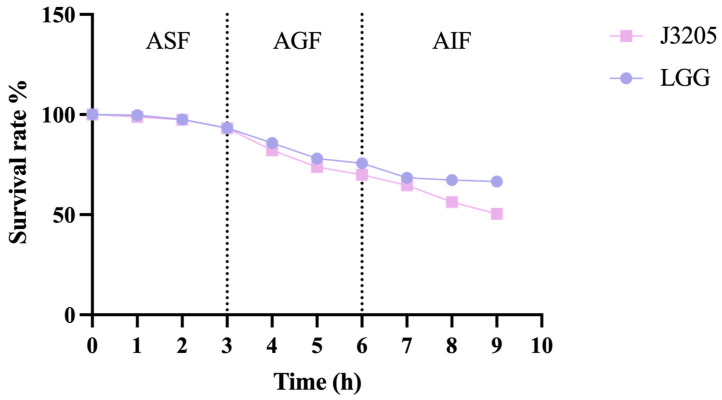
Survival rates of J3205 and LGG under artificial saliva fluid, (ASF: pH 7.0, 3 h), artificial gastric fluid (AGF: pH 2.0, 3 h), and artificial intestinal fluid (AIF: pH 8.0, 3 h).

**Figure 8 microorganisms-13-01643-f008:**
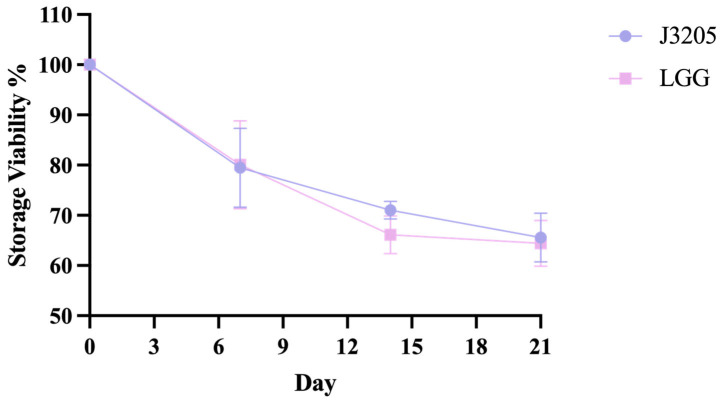
Storage viability of strains J3205 and LGG. Retention rates of *J3205* and LGG were monitored during 21 days of storage under 4 °C in freeze-dried powder. Data represent mean ± SD (*n* = 3 independent experiments). J3205 maintained significantly higher viability (65.57%) compared to LGG (64.43%) by day 21.

**Figure 9 microorganisms-13-01643-f009:**
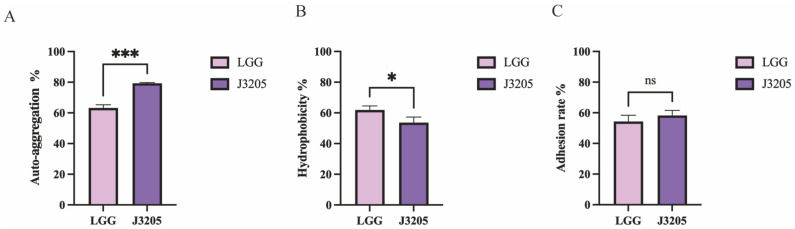
Auto-aggregation, hydrophobicity, and adhesion capacity of *L. rhamnosus* J3205 and LGG. (**A**) Auto-aggregation ability (%) after 24 h of incubation in PBS. (**B**) Hydrophobicity rate (%) measured by microbial adhesion to hydrocarbons (MATH) assay. (**C**) Adhesion rate (%) to Caco-2 intestinal epithelial cells after 2 h of incubation. Data represent mean ± SD (n = 3 independent experiments). Statistical significance: * *p* < 0.05, *** *p* < 0.001 vs. LGG; ns = not significant. J3205 showed significantly higher auto-aggregation (79.33 ± 2.15%) compared to LGG (63.20 ± 3.42%).

**Figure 10 microorganisms-13-01643-f010:**
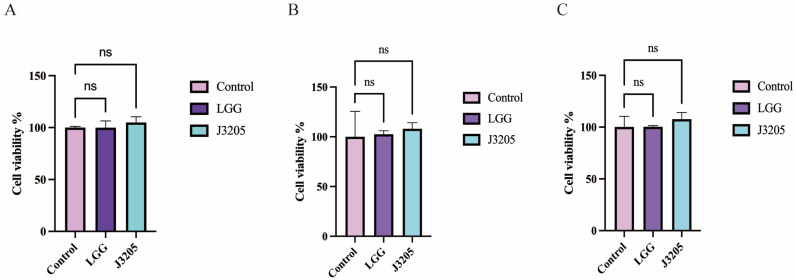
Cytotoxicity assessment of L. rhamnosus J3205 and LGG cell-free supernatants (CFS) on IEC-6 cells. Cell viability (%) after 24 h exposure to: (**A**) 5% CFS concentration (proliferative effect observed); (**B**) 10% CFS concentration; (**C**) 20% CFS concentration. Data normalized to untreated control (100% viability) and expressed as mean ± SD (*n* = 3). Significance vs. control: ns: non-significant. Both strains showed >90% viability at all tested concentrations, indicating low cytotoxicity.

**Figure 11 microorganisms-13-01643-f011:**
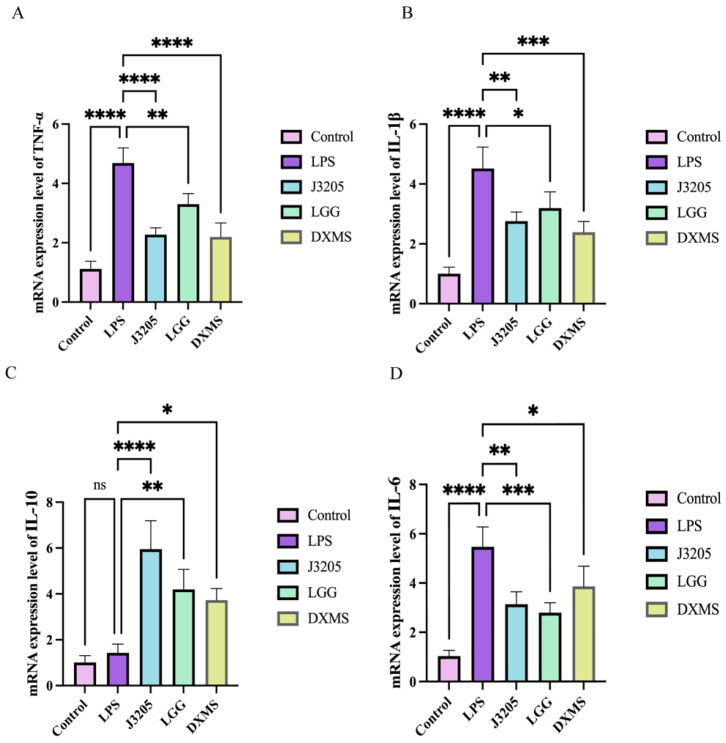
*L. rhamnosus* CFS show anti-inflammatory effects on LPS-treated Raw 264.7 macrophages. (**A**) mRNA expression level of TNF-α; (**B**) mRNA expression level of IL-1β; (**C**) mRNA expression level of IL-10; (**D**) mRNA expression level of IL-6. Cells were pre-treated with 20% bacterial supernatants (J3205 or LGG) or dexamethasone (DXMS, 50 μg/mL) before LPS (2 μg/mL) stimulation. Data are shown as mean ± SEM (*n* = 3). (ns: non-significant, * *p* < 0.05, ** *p* < 0.01, *** *p* < 0.001, **** *p* < 0.0001).

**Figure 12 microorganisms-13-01643-f012:**
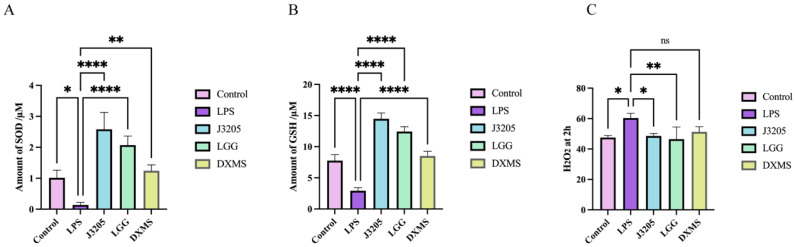
*Lacticaseibacillus* CFS shows antioxidant effects on LPS-treated Raw 264.7 macrophages. (**A**) Enzymatic activity of SOD; (**B**) amount of GSH; (**C**) generation of cellular H_2_O_2_ in 2 h. Cells were pre-treated with 20% bacterial supernatants (J3205 or LGG) or dexamethasone (DXMS, 50 μg/mL) before LPS (2 μg/mL) stimulation. Data are shown as mean ± SEM (*n* = 3). (ns: non-significant, * *p* < 0.05, ** *p* < 0.01, **** *p* < 0.0001).

**Table 1 microorganisms-13-01643-t001:** Primer sequences used for RT-qPCR.

Gene		Sequence′ (5′-3′)	Product Size (bp)	Accession Number
*β-actin*	Forward	ATGACCCAAGCCGAGAAGG	185	NM_027493
	Reverse	CGGCCAAGTCTTAGAGTTGTTG		
*tnf-α*	Forward	CCACGCTCTCTTCTGTCTACTG	169	NM_010851.2
	Reverse	ACTTGGTGGTTTGCTACGAC		
*il-10*	Forward	GGACCAGCTGGACAACATACTGCTA	80	NM_010548.2
	Reverse	CCGATAAGGCTTGGCAACCCAAGT		
*il-1β*	Forward	TTGAAAGTCCACCTCCTTACAGA	106	NM_031168.1
	Reverse	CCGGATAAAAAGAGTACGCTGG		
*il-6*	Forward	GAGTCACAGAAGGAGTGGCTAAGG	129	NM_008756
	Reverse	CGCACTAGGTTTGCCGAGTAGATCT		

**Table 2 microorganisms-13-01643-t002:** Characteristics of *L. rhamnous* J3205 genome.

Attributes	Values
Genome Size (bp)	2,882,186
G + C content (%)	46.77
5S rRNA	5
16S rRNA	5
23S rRNA	5
Plasmids	0
CRISPR number	3
tRNA	5

**Table 3 microorganisms-13-01643-t003:** Virulence factors of *L.rhamnosus* J3205 genome.

VF Gene	Virulence Factor
*galF*	UTP-glucose-1-phosphate uridylytransferase HasC
*tuf*	Adherence
*arlR*	Response regulator
*rfbB*	Immune modulation
*groEL*	Adherence

**Table 4 microorganisms-13-01643-t004:** Antibiotic resistance of the *L.rhamnosus* J3205 and LGG.

	*L. rhamnosus* J3205	*L. rhamnosus* LGG
Streptomycin	S	S
Gentamicin	S	S
Kanamycin	R	R
Tetracycline	S	S
Ampicillin	S	S
Clindamycin	I	S
Erythromycin	S	S

Notes: S: susceptibility, R: resistance, I: intermediate.

## Data Availability

The whole genome sequence data reported in this paper have been deposited in the Genome Warehouse in National Genomics Data Center, Beijing Institute of Ge- nomics, Chinese Academy of Sciences/China National Center for Bioinformation, under BioProject PRJCA038303, that are publicly accessible at https://ngdc.cncb.ac.cn/gwh, 7 April 2025.

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
