# Peer review of "In Vitro Evaluation of the Probiotic Properties and Whole Genome Sequencing of Lacticaseibacillus rhamnosus J3205 Isolated from Home-Made Fermented Sauce"

_microorganisms, 2025, doi:10.3390/microorganisms13071643_

Round 1
Reviewer 1 Report
Comments and Suggestions for Authors
The article “In vitro evaluation of the probiotic properties and whole genome sequencing of Lactobacillus rhamnosus J3205 isolated from home-made fermented sauce” describes the characterization of a potential probiotic Lactobacillus strain. The authors give broad in vitro evaluation of the strain emphasizing its probiotic potential. However, several areas require clarification, enhancement of context through citation of related literature, and significant language editing. My specific comments are as follows:
- The taxonomy of Lactobacillus rhamnosus has been revised to Lacticaseibacillus rhamnosus (check the website of International Scientific Association for Probiotics and Prebiotics: https://isappscience.org/new-names-for-important-probiotic-lactobacillus-species/). Change the name of the specie in the title and also in the article where the full term is mentioned.
- In the introduction explain the used LGG specie as control. Specify its specific ATCC or other standardized designation.
- The experiments in the study are well known experiments for probiotic characterization. To provide a broader scientific context, cite additional studies that have employed similar methodologies. For example, the following article is relevant: https://doi.org/10.1186/s12934-024-02612-w, which evaluates the hydrophobicity, aggregation and safety of potential probiotics.
- Line 52: Revise the “candidate lactic acid bacteria…” to “candidate probiotic…”. The current phrasing implies that the strain is a candidate for lactic acid production, which is misleading.
- Line 55: Improve the English of the last sentence. Currently, it reads as a list of tasks rather than a description of what was done in the study.
- Line 94: Explain what does “culture supernatant 4” mean If it refers to four biological replicates or a specific sample number, revise the sentence accordingly.
- Line 106: Specify the temperature you incubated the bacteria (e.g. room temperature, 37 °C).
- Lines 106-107: Revise the English of the sentence. Use past tense.
- Line 111: In the equation is stated “adhesion rate”, however, you evaluated bacterial viability and not adhesion. Please explain the equation or revise it appropriately.
- Revise the °C symbol everywhere throughout the text.
- Provide a clear explanation for the use of different centrifugal forces (x g) across various experiments. Are these variations necessary?
- In the material and method section there is missing information about how hydrophobicity and aggregation were assessed.
- Lines 213-215: Revise the sentence. “were” appears twice.
- Line 293: Specify the abbreviations in the table in table legend for S R I
- Line 363-364: Provide an appropriate citation to support the stated claim.
- Figure 1 and 2 are blurry. Replace them with higher-quality versions.
- Figure 9A: Clarify how an adhesion value of ~150% was obtained. If this is due to a normalization error or statistical artifact, please correct it or provide a valid explanation.

Author Response
The article “In vitro evaluation of the probiotic properties and whole genome sequencing of Lactobacillus rhamnosus J3205 isolated from home-made fermented sauce” describes the characterization of a potential probiotic Lactobacillus strain. The authors give broad in vitro evaluation of the strain emphasizing its probiotic potential. However, several areas require clarification, enhancement of context through citation of related literature, and significant language editing. My specific comments are as follows:
- The taxonomy of Lactobacillus rhamnosus has been revised to Lacticaseibacillus rhamnosus (check the website of International Scientific Association for Probiotics and Prebiotics: https://isappscience.org/new-names-for-important-probiotic-lactobacillus-species/).
Change the name of the specie in the title and also in the article where the full term is mentioned. - In the introduction explain the used LGG specie as control. Specify its specific ATCC or other standardized designation.
Response: Thank you for pointing out this issue. We have replaced all instances of "Lactobacillus rhamnosus" with "Lacticaseibacillus rhamnosus" in the text. In the introduction, we added “with L.rhamnosus LGG commercial probiotic strain (ATCC53103) as a control strain” at the end.
- The experiments in the study are well known experiments for probiotic characterization. To provide a broader scientific context, cite additional studies that have employed similar methodologies. For example, the following article is relevant: https://doi.org/10.1186/s12934-024-02612-w, which evaluates the hydrophobicity, aggregation and safety of potential probiotics.
Response: Thank you for your valuable suggestion! We’ve added the citation as reference 17.
- Line 52: Revise the “candidate lactic acid bacteria…” to “candidate probiotic…”. The current phrasing implies that the strain is a candidate for lactic acid production, which is misleading.
Response: Thank you for your suggestion! The relevant statement in the article has been revised.
- Line 55: Improve the English of the last sentence. Currently, it reads as a list of tasks rather than a description of what was done in the study.
Response: Thank you for your advice! We have revised the English writing of the last sentence. The revised version is: 2.5 g of the sauce sample was homogenized in 10 mL of 1× PBS buffer, serially diluted to 10⁻⁵, and plated on MRS agar. After 48 hours of anaerobic incubation at 37°C, isolated colonies were purified through two successive subcultures to ensure strain consistency.
- Line 94: Explain what does “culture supernatant 4” mean If it refers to four biological replicates or a specific sample number, revise the sentence accordingly.
Response: Thank you for your advice! We noticed that it is just typo error and thank you for pointing out. Revised version: culture supernatants of two strains were centrifuged (12000xg, 4 °C, 10 minutes) to remove cell debris, and then concentrated proteins were extracted using the BCA kit method.
Bacterial culture supernatants from both strains were used for proteomic analysis.
- Line 106: Specify the temperature you incubated the bacteria (e.g. room temperature, 37 °C).
Response: Thank you for your advice! We've added a one-sentence description: Bacteria was incubated anaerobically at 37 °C.
- Lines 106-107: Revise the English of the sentence. Use past tense.
Response: Thanks for pointing out the error. It is revised as: Bacterial counts were recorded every one hour.
- Line 111: In the equation is stated “adhesion rate”, however, you evaluated bacterial viability and not adhesion. Please explain the equation or revise it appropriately.
Response: Thank you for pointing this out! It is the writing typo. We changed the formula and corrected it as survival rate.
× 100%
- Revise the °C symbol everywhere throughout the text.
Response: Thank you for your advice! We’ve revised all °C symbols on line 69, 115, 132, 147, 152 175, 178, 179, 192, 328, 352 and 470 respectively.
- Provide a clear explanation for the use of different centrifugal forces (x g) across various experiments. Are these variations necessary?
Response: Thanks for your valuable suggestion. Based on the previous study protocol and experience of previous experiments, we applied different parameters for centrifugal forces in different experiments. For example, a low centrifugal force (6000 × g ) for bacteria is suitable for gentle sedimentation of fragile microorganisms to avoid cell rupture, e.g., when obtaining bacterial precipitation, we need a slightly gentler centrifugal force. Specifically, on line 124, Bacterial suspensions were centrifuged at 6000 xg for 5 minutes and resuspended by artificial gastric juice (2g/L NaCl, 3g/L pepsin, pH at 2.0), adjusting to 108 CFU/mL and incubating for 3 hours. On line 196: L. rhamnosus J3205 was centrifuged at 6000 × g for 10 minutes and washed three times with sterile PBS, then resuspended in the complete medium at a concentration of 1 × 108 CFU/mL.
Higher centrifugation forces (>10,000 × g) are used for separating cellular debris or microscopic particles, such as those used to obtain Cell free supernatant. Specifically, on line 114: culture supernatants of two strains were centrifuged (12000 × g , 4 °C, 10 minutes) to remove cell debris.
- In the material and method section there is missing information about how hydrophobicity and aggregation were assessed.
Response: Thank you for pointing this out! We feel sorry for missing the hydrophobicity and aggregation in the method part. We have carefully added these two parts in 2.5 and 2.6:
2.5 Hydrophobicity ability test
Overnight cultures were centrifuged at 5000 × g for 10 min at 4 °C. The cell pellets were washed twice with sterile phosphate-buffered saline (PBS, pH 7.2) and resuspended in PBS to an optical density of 0.6 ± 0.05 at 600 nm (A₀). 2 ml of bacterial suspension mixed vigorously with an equal volume of chloroform for 2 min and allowed to stand at room temperature for 30 min. The aqueous phase was carefully collected, and its optical density at 600 nm (A₁) was measured against PBS. Hydrophobicity percentage was calculated as:
Hydrophobicity percentage was calculated as:
× 100%
2.6 Auto-aggregation ability test
The preparation of bacterial solution is the same as hydrophobicity experiment, take 4mL of bacterial suspension, aspirate 1mL of the upper layer, the value of OD600 is A0; let it stand at room temperature for 12h, carefully aspirate 1mL of the upper layer, and measure the value of OD600, A1, the formula is:
× 100%
- Lines 213-215: Revise the sentence. “were” appears twice.
Response: Thank you! We’ve deleted the second “were”
- Line 293: Specify the abbreviations in the table in table legend for S R I
Response: Thanks for pointing this out. We have added the explanation on the bottom of the table: S: susceptibility, R: resistance, I: intermediate.
- Line 363-364: Provide an appropriate citation to support the stated claim.
Response: Thanks for pointing this out. The relative reference is provided:
Van Reenen CA, Dicks LMT. Horizontal gene transfer amongst probiotic lactic acid bacteria and other intestinal microbiota: what are the possibilities? A review. Arch Microbiol 2011;193:157–68.
- Figures 1 and 2 are blurry. Replace them with higher-quality versions.
Response: Thank you for your advice! We’ve renewed all figures with higher-quality versions and vector formats. Also, we will upload the original figures in the submission process.
- Figure 9A: Clarify how an adhesion value of ~150% was obtained. If this is due to a normalization error or statistical artifact, please correct it or provide a valid explanation.
Response: Thanks for pointing this out. There was indeed a slight problem with the calculations, and a recalculation modified the resultant graph. At 5% CFS, the cell viability is probably around 100%, indicating no inhibition of cell growth.

Reviewer 2 Report
Comments and Suggestions for Authors
Comments regarding the manuscript entitled "In vitro evaluation of the probiotic properties and whole genome sequencing of Lactobacillus rhamnosus J3205 isolated from home-made fermented sauce" (Manuscript ID: microorganisms-3623805).
The paper's subject is interesting because of its implications for human health. In addition, the paper fits with the interest of the journal Microorganisms.
In general, the manuscript is easy to follow. The introduction adequately communicates the importance and nature of the experimental problem, and the objective is well-stated. The material and methods are described clearly and concisely. Besides, the results are presented and discussed clearly. The paper contains valuable information for readers of Microorganisms and can be published in the journal after major revisions. Some specific comments are as follows:
- A thorough review for grammatical and typographical correctness, clarity, and conciseness is recommended. This would significantly enhance the document's overall readability and professionalism.
- The abstract should provide relevant quantitative information.
- Line 95: Please indicate the temperature scale.
- Please provide survival rates of J3205 and LGG strains in artificial saliva solutions.
- Lines 111-112: Please check the equation. Adhesion rate or survival rate?.
- Line 247: The words "environmentally information processing" are repeated.
- The scientific name "C. difficile" is wrongly written in the abscissa of Figure 5.
- Figure 5: Provide the p-value for ****.
- Provide the variation of total acidity as a function of time.
- Please provide information on the viability and stability of L. rhamnosus J3205 during storage, as these are technological criteria for selecting probiotic bacteria.
- The full meaning of an abbreviation should be defined at first mention in the text.
- Do the authors know the types of bacteriocins produced by L. rhamnosus J3205?

A thorough review for grammatical and typographical correctness, clarity, and conciseness is recommended. This would significantly enhance the document's overall readability and professionalism.
Author Response
Reviewer 2:
The paper's subject is interesting because of its implications for human health. In addition, the paper fits with the interest of the journal Microorganisms.
In general, the manuscript is easy to follow. The introduction adequately communicates the importance and nature of the experimental problem, and the objective is well-stated. The material and methods are described clearly and concisely. Besides, the results are presented and discussed clearly. The paper contains valuable information for readers of Microorganisms and can be published in the journal after major revisions. Some specific comments are as follows:
A thorough review for grammatical and typographical correctness, clarity, and conciseness is recommended. This would significantly enhance the document's overall readability and professionalism.
Response: We sincerely appreciate your constructive feedback. To address these concerns, we have recorrected some grammatical errors and made the sentence more concise.
The abstract should provide relevant quantitative information.
Response: Thanks for pointing this out. We’ve added quantitative data in the abstract part. The revised version:
Lacticaseibacillus rhamnosus J3205 was isolated from traditional fermented sauces and demonstrated potential probiotic properties. The strain exhibited high tolerance to simulated saliva (93.24% survival) and gastrointestinal conditions (69.95% gastric and 50.44% intestinal survival), along with strong adhesion capacity (58.25%) to intestinal epithelial cells. Safety assessments confirmed the absence of virulence and antibiotic resistance genes. Genomic analysis revealed stress-response genes and 34 insertion sequence (IS) elements, while proteomic profiling identified Pgk as a key enzyme in lactic acid production and SecY in oxidative stress resistance. Functionally, J3205 significantly reducing pro-inflammatory cytokines (TNF-α, IL-6, IL-1β) and enhancing antioxidant markers (SOD, GSH) in vitro. These results position L. rhamnosus J3205 as a promising candidate for gut-health foods, anti-inflammatory nutraceuticals, and oxidative-stress therapeutics, warranting further in vivo validation.
.Line 95: Please indicate the temperature scale.
Response: Thanks for your suggestion! We’ve changed the term “degree” to °C on line 69, 115, 132, 147, 152 175, 178, 179, 192, 328, 352 and 470 respectively.
Please provide survival rates of J3205 and LGG strains in artificial saliva solutions.
Response: Thanks for your suggestion! We refer to the article “Beneficial Properties and Evaluation of Survival in Model Systems of LAB Isolated from Oral Cavity” for the preparation method of artificial saliva. We added the artificial saliva experiment and the new plots and data are shown in Figure 7. Also, we’ve added method part on line 122-134 in section 2.3: Stability tests using Artificial Saliva Fluid (ASF), Artificial Gastric Fluid (AGF), and Artificial Intestinal Fluid (AIF) were conducted as previously described [27,28]. The bacterial suspension was centrifuged at 6000 x g for 5 minutes and then incubated with 100 mL of artificial saliva fluid (25 mM KH2PO4, 24 mM Na2HPO4, 150 mM KHCO3, 100 mM NaCl, 1.5 mM MgCl2, 25 mM citric acid, 15 mM CaCl2, pH 7.0), adjusted to 108 CFU/mL, for 3 hours. Subsequently, the bacterial suspension was resuspended in artificial gastric fluid (2 g/L NaCl, 3 g/L pepsin, pH 2.0), again adjusted to 108 CFU/mL, and incubated for another 3 hours. Following this, the bacteria were resuspended in artificial intestinal fluid (0.3% bile salt, 1 g/L trypsin in PBS, pH 8.0) for 3 hours. The bacterial count was determined hourly using the plate counting method, with the bacteria incubated anaerobically at 37 °C. After 3 hours of incubation, the bacteria were resuspended in artificial intestinal juice (0.3% bile salts, 1 g/L pancreatin, pH 8.0) and treated for an additional 3 hours. Bacterial counts were recorded at 1, 2, and 3 hours, following the same method as described above.
Lines 111-112: Please check the equation. Adhesion rate or survival rate?
Response: Thanks for pointing this out. It is survival rate. We’ve made the correction.
Line 247: The words "environmentally information processing" are repeated.
Response: Thanks for pointing this out. We’ve deleted the second one.
The revised sentence: KEGG enrichment analysis indicated that J3205 showed enriched pathways in metabolism, genetic information processing, environmental information processing, human diseases and cellular processes (Figure 4F).
The scientific name "C. difficile" is wrongly written in the abscissa of Figure 5.
Response: Thanks for pointing this out. We’ve made corrections on it.
Figure 5: Provide the p-value for ****.
Response: Thanks! We’ve added the p-value for ****: p<0.0001
Provide the variation of total acidity as a function of time.
Response: Thank you for your suggestions. However, in this article, Figure 6B indicated the variation of total acidity as a function of time for 18 hours.
Please provide information on the viability and stability of L. rhamnosus J3205 during storage, as these are technological criteria for selecting probiotic bacteria.
Thank you! We’v added the viability of L.rhamnosus J3205 on line 176-182
2.9 Viability of L.rhamnosus J3205 during storage
Inoculate the bacterial strain into MRS liquid medium and incubate anaerobically at 37°C for 18–24 h (until the late logarithmic phase, OD600 ≈ 1.0–1.2). Harvest the cells by centrifugation (4°C, 6000×g, 10 min), wash twice with sterile PBS, and resuspend in 20% skim milk powder (1:1 ratio with bacterial suspension). Dispense 300 μL aliquots into glass bottles (four bottles per strain for days 0, 7, 14, and 21). Freeze in liquid nitrogen and lyophilize. Monitor viable cell counts on day 0, 7, 14, and 21.
Results: The stability of bacterial strains during storage was evaluated over a 21-day period. Strain J3205 demonstrated superior storage viability, retaining an average of 65.57% of its initial concentration by Day 21. In contrast, LGG showed a lower retention rate of 64.43% under identical conditions (Figure 8).
The full meaning of an abbreviation should be defined at first mention in the text.
Response: Thank you! We’ve made the correction.
Do the authors know the types of bacteriocins produced by L. rhamnosus J3205?
Response: Thank you for your suggestion! This is very useful. We used the bacteriocin online software BAGEL4 to mine the genes for potential bacteriocin gene clusters, the matching bacteriocin was shown as Enterocin X β chain (Enterocin_X_chain_beta), a Class IIb two-peptide bacteriocin, usually produced by lactic acid bacteria (e.g. Enterococcus spp.). The following is a detailed breakdown of its function and properties: bit score 50.447, This bacteriocin may be associated with the inhibition of the growth of foodborne pathogens.We’ve added this part to the related result and discussion respectively.

Reviewer 3 Report
Comments and Suggestions for Authors
Abstract. It should be revised extensively. It should include background of the topic, main methods/analyses used, main results, and main conclusions with possible future works needed. In addition, potential industrial applications.
Line 31. “5-HT”. More details. Please explain.
Introduction. Please add more details regarding the significance, novelty and aim of the current work.
Lines 52-53. “traditional home-made fermented sauce widely used in Henan, China”. Please add more details. Sauce production, materials etc.
Line 111. Probably it is not adhesion rate.
The work is interesting and similar with many others studies reporting a novel probiotic candidate strain. However, it is limited by its focus on a single isolated previously. While the initial results are promising, this alone may not be sufficient to support publication. Incorporating further investigation—such as evaluating the strain’s functionality in a relevant food matrix or conducting in vivo studies—would significantly enhance the scientific value and applicability of the manuscript.
Author Response
Abstract. It should be revised extensively. It should include background of the topic, main methods/analyses used, main results, and main conclusions with possible future works needed. In addition, potential industrial applications.Response: Thank you! We’ve revised the abstract. The revised version: Lacticaseibacillus rhamnosus J3205 was isolated from traditional fermented sauces and demonstrated potential probiotic properties. The strain exhibited high tolerance to simulated saliva (93.24% survival) and gastrointestinal conditions (69.95% gastric and 50.44% intestinal survival), along with strong adhesion capacity (58.25%) to intestinal epithelial cells. Safety assessments confirmed the absence of virulence and antibiotic resistance genes. Genomic analysis revealed stress-response genes and 34 insertion sequence (IS) elements, while proteomic profiling identified Pgk as a key enzyme in lactic acid production and SecY in oxidative stress resistance. Functionally, J3205 significantly reducing pro-inflammatory cytokines (TNF-α, IL-6, IL-1β) and enhancing antioxidant markers (SOD, GSH) in vitro. These results position L. rhamnosus J3205 as a promising candidate for gut-health foods, anti-inflammatory nutraceuticals, and oxidative-stress therapeutics, warranting further in vivo validation.
Line 31. “5-HT”. More details. Please explain.
Response: Thank you! We’ve edited our manuscript. Since the introduction section was designed to focus on J3205, and J3205 does not have any neuro-related functions that are the primary focus of this study, we have downplayed the information regarding 5-HT, mentioning it only briefly: Additionally, certain probiotics may influence neurological function through the gut-brain axis.
Introduction. Please add more details regarding the significance, novelty and aim of the current work.
Response: Thanks for pointing this out. We’ve added more details of the aim, novelty, and significance of the work: This study aims to isolate and identify potential probiotic strains from traditional family-fermented sauces, and to evaluate their benefits and safety for human applications. The commercially available probiotic strain L. rhamnosus GG (LGG, ATCC 53103) was used as a control. The results revealed that this strain possesses excellent potential probiotic properties, particularly in alleviating oxidative stress and inflammation, suggesting its potential therapeutic value in treating gut-related and metabolic disorders.
Lines 52-53. “traditional home-made fermented sauce widely used in Henan, China”. Please add more details. Sauce production, materials etc.
Response: Thank you for your suggestion! This is really a writing issue, the sauce is a homemade sauce originally fermented from soybeans. (Line 63 in new version revised manuscript)
Line 111. Probably it is not adhesion rate.Response: Thanks for pointing this out! We’ve made the correction and recorrected it as survival rate.
The work is interesting and similar with many others studies reporting a novel probiotic candidate strain. However, it is limited by its focus on a single isolated previously. While the initial results are promising, this alone may not be sufficient to support publication. Incorporating further investigation—such as evaluating the strain’s functionality in a relevant food matrix or conducting in vivo studies—would significantly enhance the scientific value and applicability of the manuscript.
Response: I would like to express my sincere regret that this article did not meet your expectations with regard to scientific and academic rigour. A substantial body of research is required to support the screening, application and mechanistic investigation of a potential probiotic strain. In this manuscript, we conducted a biological characterization of a newly isolated candidate strain. This was not a comprehensive characterization, but it revealed the characteristics of the strain in a more comprehensive way, laying the foundation for subsequent in vivo functional studies. It is hoped that the results of subsequent studies will be shared with all researchers in the form of published articles. I would like to express my gratitude once more for your pertinent and valuable comments.

Round 2
Reviewer 1 Report
Comments and Suggestions for Authors
The article was significantly improved and it is almost ready for publication. Two minor revision suggestions:
- Revise subsection 2.9. As currently written, it is more like commends and not how the protocol was conducted. Revise the English and use proper past tenses.
- In the comments you explained why you used different centrifugal forces, but nowhere in the discussion section you state the reasons. It would be good for the readers to understand the necessity of different centrifuge parameters. Therefore, I suggest to add one or two phrases in the discussion to clarify this point.
Author Response
Response to the reviewers’ comments on Microorganisms-3623805
Dear Reviewers,
First of all, we would like to thank you for your time, constructive critiques, and valuable suggestions. We did our best to address your comments on our revised manuscript. Below, we give a detailed response to each of your comments.
Your input contributed to a significant improvement of the paper and to our proposed concepts as well.
All changes made to the previously submitted document have been marked in blue to highlight them for you.
Sincerely yours,
- Yiming and L. Chang
Response to Reviewer 1
- Revise subsection 2.9. As currently written, it is more like commends and not how the protocol was conducted. Revise the English and use proper past tenses.
Response: Thank you for your suggestion! The revised version is as follows (Line 175 to 183):
2.9 Viability of L.rhamnosus J3205 during storage
The bacterial strain was inoculated into MRS liquid medium and incubated anaerobically at 37 °C for 18–24 h (late logarithmic phase, OD600≈ 1.0–1.2). Cells were harvested by centrifugation (4 °C, 6000 × g, 10 min), washed twice with sterile PBS, and resuspended in 20% (w/v) skim milk powder at a 1:1 ratio (bacterial suspension to skim milk). Aliquots (300 μL) were dispensed into sterile glass bottles (four bottles per strain, corresponding to storage days 0, 7, 14, and 21). The samples were flash-frozen in liquid nitrogen and lyophilized. Viable cell counts were determined on days 0, 7, 14, and 21 by standard plate counting on MRS agar.
- In the comments you explained why you used different centrifugal forces, but nowhere in the discussion section you state the reasons. It would be good for the readers to understand the necessity of different centrifuge parameters. Therefore, I suggest to add one or two phrases in the discussion to clarify this point.
Response: Thank you for your suggestion! We added the following to the first paragraph of our discussion (Line 410 to 414): In this study, different centrifugal forces were used to separate bacterial pellets and supernatants (6,000 ×g was used to collect bacterial pellets, and 12,000 ×g was used to clarify supernatants). Low-speed centrifugation was used to preserve bacteria and prevent damage, while cells in the supernatant were typically collected by high-speed centrifugation as previously described [33].
Reviewer 2 Report
Comments and Suggestions for Authors
Comments regarding the manuscript entitled " In vitro evaluation of the probiotic properties and whole genome sequencing of Lacticaseibacillus rhamnosus J3205 isolated from home-made fermented sauce" (Manuscript ID: microorganisms-3623805)
The authors considered most of my comments and submitted an improved version of their manuscript, so the revised manuscript may be considered to be accepted for publication.

Author Response
Response to Reviewer 2
Comments regarding the manuscript entitled " In vitro evaluation of the probiotic properties and whole genome sequencing of Lacticaseibacillus rhamnosus J3205 isolated from home-made fermented sauce" (Manuscript ID: microorganisms-3623805)
The authors considered most of my comments and submitted an improved version of their manuscript, so the revised manuscript may be considered to be accepted for publication.
Response:
Dear Reviewer,
We sincerely appreciate your time and constructive comments on our manuscript (ID: microorganisms-3623805). We are grateful for your thorough review and are pleased that our revisions addressed your concerns. Your insightful suggestions have significantly improved the quality of our work.
Thank you for your positive assessment and recommendation for acceptance. We look forward to contributing this research to Microorganisms.
Sincerely yours,
- Yiming and L. Chang